# Complex intervention based on protective factors to improve resilience for gastric cancer patients: Mixed-methods process evaluation protocol

Xia Zhao[1,2], Min Fan[1], Yanan Ding[3], Yuqin Pan[4], Xinqiong Zhang[1]*

1 School of Nursing, Anhui Medical University, Hefei, Anhui, China, 2 The First Affiliated Hospital of Anhui Medical University, Hefei, Anhui, China, 3 Anhui Sanlian University, Hefei, Anhui, China, 4 Department of Oncology, The First Affiliated Hospital of Anhui Medical University, Hefei, Anhui, China

* hixqzhang@163.com

## Abstract

### Background

Gastric cancer represents a global health burden, with patients often experiencing significant psychological distress due to various treatments. Resilience, the ability to adapt positively to challenges, is crucial for the mental and physical well-being of gastric cancer patients. Despite its importance, there are still gaps in research aimed at enhancing resilience. This paper presents a process evaluation protocol embedded within a complex intervention aimed at enhancing resilience in gastric cancer patients during chemotherapy. The protocol aims to understand the implementation process, mechanisms, influencing factors, and outcomes.

### Methods

The process evaluation will use a mixed-methods approach. The Consolidated Framework for Implementation Research (CFIR) and the RE-AIM framework were chosen to qualitatively identify the factors that influence intervention implementation and the outcomes of implementation, including the reach, efficacy, adoption and maintenance. The implementation process will be assessed based on the UK Medical Research Council's guidance on evaluating complex interventions. The fidelity, dose and adherence will be analysed through the goal achievement log book, and the mechanisms of intervention implementation will be assessed using parallel latent growth curve model with quantitative information. Semi-structured interviews will be conducted with intervention staff and participants, data from the goal achievement log book will be analyzed, and data from the primary and secondary outcome indicators will be analyzed for parallel latent growth curve parameters.

**Data availability statement:** No datasets were generated or analysed during the current study. All relevant data from this study will be made available upon study completion.

**Funding:** This paper was supported by the Humanities and Social Sciences Research Planning Fund of Ministry of Education of China (grant number: 17YJAZH126) and the Anhui Provincial Department of Education (grant number: yjs20210299).The funders had no role in study design, data collection and analysis, decision to publish, or preparation of the manuscript.

**Competing interests:** The authors have declared that no competing interests exist.

## Discussion

This study outlines a process evaluation protocol for a complex intervention. The protocol aims to further validate the effectiveness of the intervention and guide the development of more comprehensive resilience strategies, thus promoting broader application among other chronic disease patient populations.

## Trial registration

The trial has been prospectively registered on 11 July 2023; ChiCTR2300073466.

## Introduction

According to the latest statistics from the International Agency for Research on Cancer (IARC) [1], gastric cancer (GC) ranked fifth globally in terms of both new cases and deaths in 2022, highlighting its significant burden on global public health. As a fatal disease [2–3], GC patients face various treatment modalities [4], including surgery and chemotherapy, and suffer severe psychological distress such as depression, anxiety, and psychological challenges. Study has shown that chemotherapy is an important therapeutic modality for GC patients, but it is also a significant source of physical and psychological stress [5]. In a study on the risk factors for the efficacy of chemotherapy in advanced GC patients conducted by Zhang et al. [6], psychological distress was identified as an important factor that affects patients' physical symptoms and treatment outcomes. Research [7] has shown that patients who adopt positive coping strategies are more likely to regulate, adapt, and grow with a positive and optimistic mindset when facing physical and psychological challenges. With the development of positive psychology, the study of resilience, focusing on individuals' positive psychological responses, has become a research hotspot [8–9]. Resilience is a process in which an individual, facing internal and external stress and adversity, activates latent internal cognitions, abilities, or psychological qualities and employs internal and external resources to actively repair and adjust, thereby acquiring the ability, process, or outcome of moving toward positive goals [10]. Studies have demonstrated [11] a close correlation between resilience in GC patients and their likelihood of survival; higher resilience is associated with greater confidence in facing the disease. The study by Zhang et al. found that individuals with higher resilience are usually able to evaluate themselves more positively and are more proactive in problem-solving and seeking help [12]. A systematic review of resilience in adult cancer patients by Tamura et al. revealed that resilience is closely related to alleviating anxiety and depression in cancer patients and improving their quality of life. Numerous studies have explored interventions that promote psychological resilience. A systematic review [13] has shown that psychological health interventions targeting resilience can effectively improve symptoms and reduce suffering in cancer patients. However, existing interventions mostly rely on general psychological therapies such as attention and interpretation therapy and cognitive interventions, lacking targeted interventions focusing on resilience itself. In a large-sample longitudinal study

spanning six months [14], our research team investigated psychological resilience among chemotherapy-treated GC patients, identifying social function, family function, self-efficacy, and hope as protective factors contributing to resilience in these patients. Based on these findings, our team developed an intervention program centered around protective factors of psychological resilience aimed at enhancing resilience during chemotherapy for GC patients. A pilot study demonstrated [15] feasibility and acceptability of this intervention. To further validate its effectiveness and refine intervention strategies, a large-sample formal trial study is currently underway.

The large-sample chemotherapy-phase GC patient resilience intervention study (referred to as "resilience intervention" below) is a single-blind, quasi-experimental study. Due to ethical and practical considerations, a non-concurrent quasi-experimental design was chosen. Given the nature of the intervention, blinding of researchers and patients was not feasible; however, single-blinding was implemented for outcome assessors. Before the implementation of the intervention, the intervention developers trained the research team members and clinical nurses, all of whom held psychological counseling certification. They provided a detailed introduction to the intervention plan and emphasized important considerations for its implementation, ensuring that they had a clear understanding of the intervention measures and could execute them consistently, thereby ensuring the standardization and homogeneity of the intervention implementation.

Convenience sampling was used to select 196 chemotherapy-phase GC patients treated at a tertiary hospital in China as study participants, all meeting inclusion and exclusion criteria. Inclusion criteria were as follows (1) patients with a histopathological diagnosis of gastric cancer, (2) age ≥ 18 years, (3) post-gastrectomy chemotherapy planned to be implemented at the study hospital, (4) patients with a permissible medical condition who provided informed consent and were able to cooperate, (5) patients with an estimated survival of more than 6 months and eligible for follow-up, (6) patients who completed the resilience intervention. Exclusion criteria included (1) patients with communication or comprehension barriers, (2) patients with other severe comorbidities, (3) patients with a history of psychological disorders or psychiatric illness, (4) patients participating in other clinical studies. Dropout criteria were applied to patients who did not complete the entire resilience intervention for any reason. Following patient informed consent, those admitted from August 2023 to March 2024 were assigned to the control group, while those admitted from April 2024 to December 2024 were assigned to the intervention group. At baseline, outcome measures were collected for both groups of patients. The primary outcome measure was the level of resilience, assessed using the 14-Item Resilience Scale [16]. This scale includes two dimensions: personal competence and acceptance of self and life, and uses a Likert 7-point rating scale ranging from 1 (not at all) to 7 (completely), with total scores ranging from 14 to 98 points; higher scores indicate better resilience. Secondary outcome measures included: (1) hope level, assessed using the Herth Hope Index [17], which includes three dimensions: positive attitude towards reality and the future, positive actions taken, and close relationships with others, and uses a Likert 4-point rating scale with total scores ranging from 12 to 48 points (low hope: 12–23 points, moderate hope: 24–35 points, high hope: 36–48 points); (2) negative emotions (anxiety and depression), assessed using the Self-Rating Anxiety Scale and the Self-Rating Depression Scale [18]; (3) family function, assessed using the Chinese Version of the Family Hardiness Index [19], which includes three dimensions: responsibility, control, and challenge, and uses a Likert 4-point rating scale ranging from "strongly disagree" to "strongly agree" (1–4 points), with higher scores indicating better family hardiness; (4) social function, assessed using the Social Support Rating Scale [20], which includes three dimensions: objective support, subjective support, and utilization of social support, with higher total scores indicating greater social support; and (5) biochemical indicators, including C-reactive protein, cortisol, and interleukin levels, obtained through the hospital database.

After baseline data collection, the control group received standard psychological care measures (see Table 1). The intervention group underwent the resilience intervention, administered by the research team and clinical nurses. The intervention spanned two months with six sessions, each lasting approximately 30–50 minutes. A blended approach combining online and in-person methods was utilized: researchers conducted face-to-face sessions with patients during hospitalization, supplemented by online sessions during chemotherapy intervals. Each session included a video segment accessible via QR code for patient viewing or facilitated by intervention staff. These videos aimed to enhance patients'



**Table 1. Control group psychological nursing content.**

| Care measure | Care content |
|---|---|
| Nursing assessment | Assess psychological, familial, and social support, mental status, and disease perception. |
| Nursing interventions | ① Condition monitoring: Closely monitor the patient's psychological changes, communicate promptly with the patient's family, and prevent adverse events; ② Symptomatic nursing: Provide targeted guidance to the patient to stabilize emotions; ③ Medication care: Implement nursing interventions tailored to different medications to alleviate the patient's anxiety and tension. |
| Health education | ① At Admission: Engage in proactive communication with the patient, introduce the responsible physician and nursing staff, and facilitate the patient's adaptation to the ward environment; ② During Hospitalization: Maintain a positive attitude, actively engage in treatment, clarify patient doubts, and offer support as needed; ③ At Discharge: Provide guidance on self-management of chemotherapy side effects and emphasize the importance of regular follow-up appointments and monitoring of relevant health indicators. |

understanding of the intervention process and content. Table 2 provides an overview of the intervention content; detailed intervention information can be found in the pilot study [15]. Following the intervention period, outcome measures were again assessed in groups to evaluate the intervention's effectiveness based on changes in primary and secondary outcome indicators.

Resilience intervention is complex due to its multifaceted intervention needs across cognitive, emotional, behavioral, and social support dimensions, as well as the dynamic characteristics of individual differences, environmental factors, and psychological states. Therefore, resilience intervention is complex. Complex interventions share two common features: the complexity of the intervention itself and the complexity of its pathways [21]. Process evaluation is a crucial component in designing and testing complex interventions, equally important as outcome evaluation [22]. Process evaluation provides scientific evidence and empirical support for the successful implementation and continuous improvement of intervention measures from many aspects [21]. Therefore, resilience intervention needs to employ process evaluation to identify the mechanisms of each intervention component and the underlying logic behind the outcomes. This study elaborates on the process evaluation protocol for resilience intervention. The protocol aims to understand the implementation process and influencing factors, complementing the outcome evaluation by illustrating the impact of process evaluation on the results, thereby providing insights for developing more refined intervention strategies and broader application.

Specific objectives are to:

1. Assessing the reach, efficacy, adoption, maintenance, fidelity, dose, and adherence of the intervention implementation.

2. Exploring the mechanisms of the intervention measures and the impact of process on intervention results.

3. Exploring the facilitating and barrier factors of intervention implementation from the stakeholders' perspective, as well as their views on the intervention.

## Materials and methods

### Design

This study employs a mixed-methods approach integrating qualitative and quantitative data, focusing on the process, mechanisms, factors, and outcomes of intervention under effectiveness trial context.

**Table 2. Overview of the resilience intervention.**

| Intervention sessions | Intervention themes | Intervention content | Intervention form | Intervention staff |
|---|---|---|---|---|
| No.1 | Fostering trust | 1. Establish a good relationship<br>2. Introduce the research objectives<br>3. Show videos and distribute the goal achievement log book | Face-to-face | Team members & clinical nurses |
| No.2 | Goal setting | Guide the setting of appropriate goals | Face-to-face | Team members & clinical nurses |
| No.3 | Family function | 1. Guide review of family support<br>2. Watch intervention videos<br>3. Assign homework | WeChat | Team members |
| No.4 | Self-efficacy | 1. Role model inspiration<br>2. Encourage positive thinking<br>3. Assign homework | Face-to-face | Team members & clinical nurses |
| No.5 | Social support | 1. Provide social and peer support<br>2. Assign homework | WeChat | Team members |
| No.6 | Self-efficacy strengthening | 1. Guide reverse thinking<br>2. Show videos on emotional regulation<br>3. Assign homework | Face-to-face | Team members & clinical nurses |

Process evaluation [21] follows the guidance of the Medical Research Council (MRC) framework. This framework directs several stages of complex intervention: development of intervention measures, feasibility assessment in the pilot phase, the implementation, and evaluation in the formal trial stage. During the formal trial stage, MRC emphasizes assessing the quantity and quality of intervention delivery content to provide greater confidence in effectiveness conclusions. At the beginning of the intervention, the intervention staff will distribute goal achievement log books (S1 File) to the patients in the intervention group to record fidelity, dose, and adherence of implementation processes. The delivery of the intervention will be quantitatively assessed through data analysis after its completion.

For intervention mechanisms, parallel latent growth curves [23] with quantitative data will be employed to explain how the intervention produces effects and the underlying processes of these effects. [24] Parallel Latent Growth Curves is commonly used in psychology, social sciences, medicine, and other fields to explore potential trends in variable changes over time and whether differences exist between individuals or groups in these changes. In intervention research, Parallel Latent Growth Curves can assess the intervention's impact on changes in outcome variables over time.

Additionally, the Consolidated Framework for Implementation Research (CFIR) and the RE-AIM framework will be used to guide the determination of influencing factors and outcomes. CFIR [25] is a meta-theoretical framework for implementation that comprehensively categorizes specific constructs related to intervention. These include intervention characteristics, inner and outer setting, characteristics of individuals, and process. CFIR will be used to guide the exploration of factors influencing the implementation process.The RE-AIM framework [26] will further guide our evaluation of the implementation outcomes, including (1) Reach: the proportion of the target population that participates in the intervention, (2) Effectiveness: the impact of the intervention, (3) Adoption: the willingness of the target population to adopt the intervention, (4) Implementation: the implementation of the intervention in real-world setting, (5) Maintenance: the sustainability of the intervention after its implementation. Semi-structured interviews will be used to collect qualitative data for identifying influencing factors and outcomes.

The study description follows the SPIRIT checklist(S2 File), with detailed enrollment, intervention, and assessment timelines provided in Fig 1.

| STUDY PERIOD | Before intervention | | | Intervention period | | | | | Post-intervention |
|---|---|---|---|---|---|---|---|---|---|
| TIMEPOINT** | Enrolment | Allocation | Baseline | First chemotherapy phase | Interval phase | Second chemotherapy phase | Interval phase | Third chemotherapy phase | Post-intervention |
| **ENROLMENT:** | | | | | | | | | |
| **Eligibility screen** | X | | | | | | | | |
| **Informed consent** | X | | | | | | | | |
| **Allocation** | | X | | | | | | | |
| **INTERVENTIONS:** | | | | | | | | | |
| **Control group：** standard psychological care | | | | X | | X | | X | |
| **Intervention group:** | | | | | | | | | |
| Fostering trust | | | | X | | | | | |
| Goal setting | | | | X | | | | | |
| Family function | | | | | X | | | | |
| Self-efficacy | | | | | | X | | | |
| Social support | | | | | | | X | | |
| Self-efficacy strengthening | | | | | | | | X | |
| **ASSESSMENTS:** | | | | | | | | | |
| **Primary outcome：** levels of resilience | | | X | X | | X | | X | |
| **Secondary outcomes：** levels of hope; levels of negativ emotions; levels of family function; levels of social function; levels of C-reactive protein, cortisol,  interleukin | | | X | X | | X | | X | |
| **PROCESS EVALUATION:** | | | | | | | | | |
| **Patients and significant others** | | | | | | | | | |
| Semi-structured interview | | | | | | | | | X |
| Parallel latent growth curve model analysis | | | | | | | | | X |
| Goal achievement log book analysis | | | | | | | | | X |
| **Intervention staff** | | | | | | | | | |
| Semi-structured interview | | | | | | | | | X |

**Fig 1. SPIRIT schedule of enrolment, data collection and assessments.**

## Logic model

The Implementation Research Logic Model (IRLM) [27] is being employed to design the process evaluation (Fig 2). The IRLM provides a method to describe the complex relationships between key elements of implementation research and practice, enhancing rigor and replicability in research and implementation practice. The IRLM specifies the following core elements of implementation research: (1) determinants of implementation, (2) implementation strategies, (3) mechanisms, (4) outcomes. The IRLM offers a systematic and comprehensive approach to integrate resources from various stakeholders. This model will be used to present and explain our findings and to test whether potential causal hypotheses are correct.

## Data collection and analysis

[28] Limbani et al suggested that collecting process evaluation data before outcome data are available could aid in exploring implementation process without inadvertently influencing researcher or patient behaviors in the study. As this

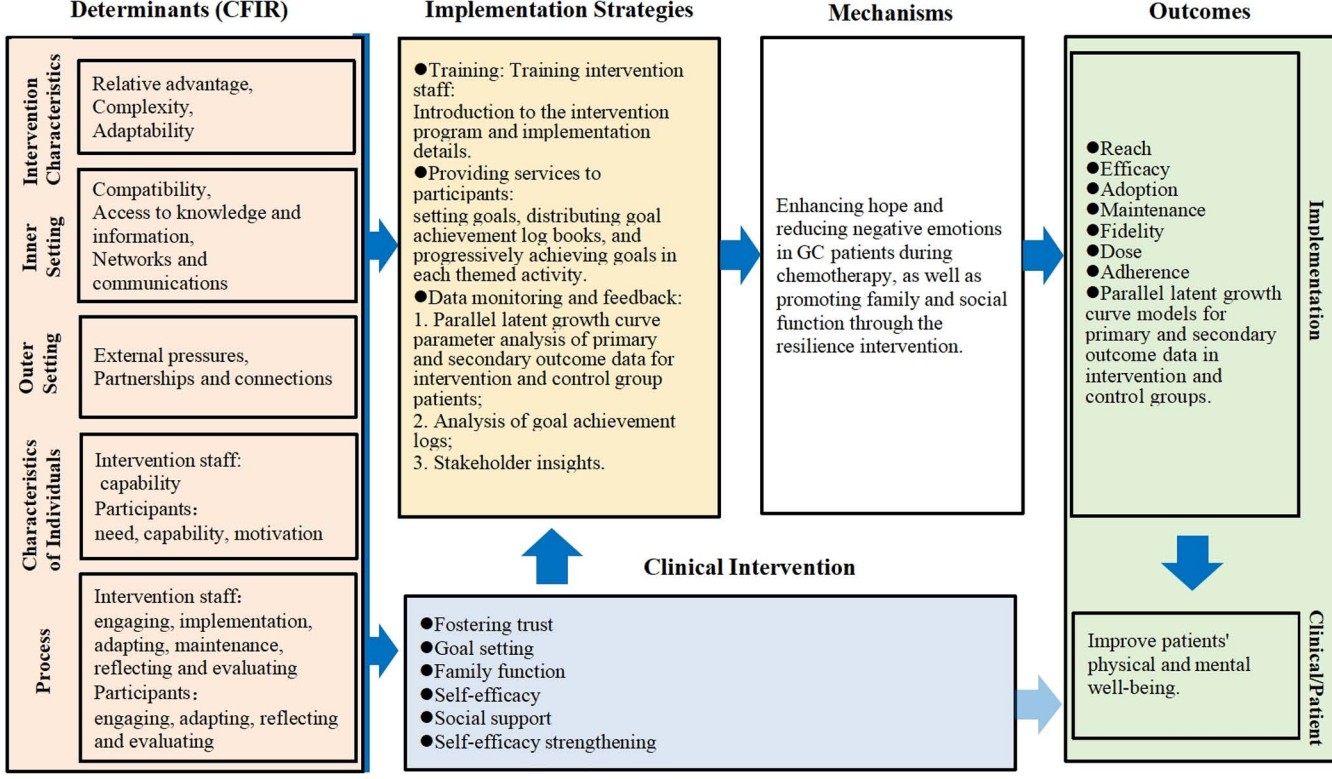

**Fig 2. The logical model of process evaluation.** GC, gastric cancer.

protocol is embedded within a large-sample formal trial, during the effectiveness research phase, to ensure the accuracy and reliability of study results, goal achievement log books will be collected and qualitative data will be gathered through semi-structured interviews after the intervention has concluded. Quantitative data analysis will also be conducted. Details regarding outcome data collection methods and contents are outlined in Table 3.

## Qualitative data collection and analysis

**Semi-structured interviews with intervention staff and participants.** Firstly, semi-structured interviews will be conducted with intervention staff to identify factors influencing intervention implementation and gather their perspectives on the intervention. The interview guide will be developed and refined using the question repository provided on the CFIR website (https://cfirguide.org/constructs/), which includes intervention characteristics (relative advantage, complexity, adaptability), inner setting (compatibility, access to knowledge and information, networks and communications), outer setting (external pressures, partnerships and connections), characteristics of individuals (capability), processes (engaging, implementation, adapting, maintenance, reflecting and evaluating).

Next, interviews will be conducted with intervention participants with the aim to determine from their perspective: (1) factors influencing intervention implementation and their views on the intervention; (2) reach; (3) efficacy; (4) adoption; (5) maintenance. The interview guide development will be based on the theoretical approaches mentioned above. Determinants will be based on the CFIR framework, including intervention characteristics (relative advantage, complexity, adaptability), inner setting (compatibility, space, access to knowledge and information), outer setting (external pressures, partnerships and connections), characteristics of individuals (need, capability, motivation), and process (engaging,

**Table 3. Process evaluation outcome data collection methods.**

| Outcome | Description | Method of evaluation |
| --- | --- | --- |
| Reach | The proportion of the target population that participates in the intervention. | Semi-structured interview |
| Efficacy | Effect of intervention measures under ideal conditions. | Semi-structured interview |
| Adoption | Patient acceptance of the intervention. | Semi-structured interview |
| Maintenance | Possibility and necessity of sustained intervention use. | Semi-structured interview |
| Fidelity | The degree to which an intervention is delivered as intended. | Goal achievement log data analysis |
| Dose | Amount of intervention delivered to participants. | Goal achievement log data analysis |
| Adherence | The degree to which participants conform to the intervention protocol. | Goal achievement log data analysis |
| Parallel latent growth curve models for primary and secondary outcome data in intervention and control groups | Parallel latent growth curves of primary and secondary outcomes for intervention and control group patients over time, along with the differences between the curves. | Analyse intervention outcome data for parallel latent growth curve parameters |

adapting, reflecting and evaluating). Reach, efficacy, adoption, and maintenance will be determined based on the RE-AIM framework.

Interview participants will be purposively sampled from the intervention group patients, following the principle of maximum variation sampling, selecting patients with significant differences in age, education level, occupation, religious beliefs, etc, until data saturation is reached, meaning no new themes emerge.

All interviews will be conducted in private, quiet hospital interview rooms after informing participants about the purpose and methods of the interview and obtaining their informed consent.The interviews will be conducted in person, one-on-one, by external staff who are independent of the intervention and experienced in qualitative data collection. Each interview will last 30 minutes or longer, recorded via audio recording, anonymized, and transcribed verbatim. Data organization and analysis will be conducted using NVivo 11.0 software, and the Colaizzi phenomenological 7-step analysis method will be used to organize the interview recordings. The domains and content of the interview guide are provided in S3 File.

### Quantitative data collection and analysis

**Parallel latent growth curve model analysis.** In addition to qualitative data, quantitative data will be collected with the aim of exploring the mechanisms of the intervention and the process of change, as well as the intervention's impact on outcomes. An intention-to-treat analysis will be employed to provide a more comprehensive and robust assessment of the intervention effects, with multiple imputations for missing values conducted using R software. Parallel latent growth curve models [24] will be employed, which allow simultaneous analysis of multiple outcome variables and account for individual differences. The primary outcome (resilience level) and secondary outcomes (hope level, anxiety and depression levels, family function, social function, C-reactive protein, cortisol, interleukins) for both the intervention and control groups will be collected one day before the end of each chemotherapy hospitalization. These data will be collected three times, with each collection separated by one chemotherapy cycle. Data analysis will be conducted using R software. Model specifications will define growth factors for each outcome variable, assuming linear growth trajectories captured by time variables. Random effects will include random intercepts and slopes to allow for variability in growth trajectories between individuals. Maximum Likelihood Estimation (MLE) will be used to estimate model parameters and evaluate model fit using indices including Comparative Fit Index (CFI), Tucker-Lewis Index (TLI), and Root Mean Square Error of Approximation (RMSEA). The significance level will be set at $p < 0.05$.

**Goal achievement log book analysis.** After the final intervention session, goal achievement log books will be collected from patients to assess fidelity, dose, and adherence of implementation, aiming to understand the delivery of the intervention. Firstly, the text will undergo cleaning, which includes removing irrelevant content, formatting, and preparing the text for analysis. Secondly, the content of the logs will be carefully reviewed: fidelity will be reflected by the percentage of intervention plan completed by participants, dose will be assessed by the duration of participant engagement in the intervention, and adherence will be evaluated by the percentage of intervention tasks completed by participants. Finally, descriptive statistics will be used to describe the central tendency and variability of fidelity, dose, and adherence. This process will be conducted by external staff independent of the intervention.

## Quality control

### Study design phase.

(1)  A mixed-methods approach will be used, integrating quantitative and qualitative data to comprehensively explore the implementation process of the intervention, ensuring the accuracy and comprehensiveness of the results.

(2)  The mixed-methods study will be guided by the MRC guidelines, the RE-AIM framework, the CFIR framework, and the parallel latent growth curve model to ensure the scientific rigor and reliability of the process evaluation.

### Intervention implementation phase.

(1)  Before the intervention, the intervention developers will train the research team members and clinical nurses, providing a detailed introduction to the intervention plan and emphasizing important considerations for its implementation, ensuring that they have a clear understanding of the intervention measures and can execute them consistently, thereby ensuring the standardization and homogeneity of the intervention implementation.

(2)  Intervention staff: All intervention implementers will hold psychological counseling certification. The research team will hold weekly discussion meetings to report on the progress of the intervention and collectively discuss and resolve any issues that arise, further ensuring the quality of the intervention implementation.

(3)  Intervention subjects and their families: The purpose, significance, and potential benefits of the intervention will be explained in detail to the patients and their families. Their consent will be obtained, and they will be asked to sign an informed consent form. Family members will be involved throughout the process and encouraged to provide supervision, enhancing the enthusiasm and compliance of the study subjects.

(4)  Intervention implementation: The intervention will be delivered through face-to-face communication and WeChat for real-time supervision of the patients, with timely resolution of any intervention-related issues. The intervention plan will be dynamically adjusted to ensure high-quality implementation.

### Data collection phase.

(1)  Quantitative data: The final data will be formed through a combination of self-assessment by patients via the goal achievement log book and assessments recorded by researchers, providing a multi-perspective reflection of the intervention implementation. Qualitative data: The interview guide will be developed based on the question repository provided by the CFIR website and refined after a pilot interview. The data collection tools will be scientifically reliable. Semi-structured interviews will be conducted by experienced interviewers to facilitate comprehensive expression of patients' thoughts and ensure the relative objectivity of the interview data.

(2)  Throughout the data collection process, investigators will be well-versed in the content, scoring methods, and usage notes of each scale, using a unified set of instructions to personally guide patients in filling out the forms, reducing

information bias. If patients are unable to complete the questionnaire independently, researchers will read out the questionnaire content and fill it in according to the patients' responses, ensuring the authenticity of the questionnaire completion. After the questionnaires are collected, they will be promptly checked for any missing items and completed as necessary to ensure data integrity.

(3) Data processing will be conducted by professionals who will double-check and enter the data to ensure the objectivity and accuracy of data processing.

### Data analysis phase

Data processing and analysis will be conducted by external personnel independent of the intervention implementation process to achieve single-blinding and reduce bias in result analysis.

### Ethics statement

This study has received ethical approval on July 11th, 2023, from the Ethics Committee of Anhui Medical University, China (ethics number: 84230068). All participants in the study signed an informed consent form after being informed of the study's purpose, and submitted it on paper. Participation in the process evaluation was entirely voluntary, and all participants had the right to refuse to participate and withdraw from the study at any time.

### Discussion

This study outlines a process evaluation protocol for a complex intervention aimed at enhancing resilience in GC patients during chemotherapy. The protocol explores the implementation process of the intervention to complement and substantiate the results of the effectiveness evaluation. It also revises and refines the intervention program to form a more comprehensive intervention strategy, thereby facilitating the broader application of the intervention among other chronic disease patient populations. Guided by the MRC guidelines [21], CFIR framework [25], and RE-AIM framework [26], the protocol ensures the scientific rigor and completeness of the study. The MRC guidelines are widely used for the development and evaluation of complex interventions. A study by Collet et al. [29] showed that the CFIR framework is a widely used framework to identify facilitators and barriers to the successful implementation of healthcare interventions. A systematic review of RE-AIM utilization by D'Lima et al. [30] indicated that the RE-AIM framework is one of the most widely used frameworks for planning and evaluating the implementation of complex interventions and is now widely used in qualitative and mixed-methods applications. These frameworks will help to fully understand the implementation process of the intervention, the context factors related to the effectiveness of the intervention, and to obtain information to promote better application and dissemination of the intervention among GC patients. Since this intervention is based on resilience itself and its protective factors, and the feasibility and acceptability of the intervention have been demonstrated in GC patients in previous studies, this protocol not only provides strong evidence for the broader application of the intervention in the future but also lays a solid foundation for conducting resilience enhancement studies in other cancer patients and even other chronic disease patients. Our research results will help to develop evidence-based guidelines and provide a scientific basis for health policy and practice. By enhancing patients' resilience, the study aims to improve the overall quality of care for patients and thus bring positive impacts to the healthcare system.

The protocol employs an embedded design, using the IRLM to better understand the core implementation elements of the resilience intervention, including determinants, implementation strategies, mechanisms, outcomes, and potential causal relationships between elements. A mixed-methods design is used to integrate qualitative and quantitative data to comprehensively reflect the implementation of the intervention. A review of process evaluations in randomized controlled trials by French et al. [31] found that of the 31 studies included, 17 used process evaluations, of which nine were quantitative and eight were qualitative. A scoping review of implementation plans for telehealth services by Rangachari et al. [32]

showed that most studies (77%) were qualitative or mixed-methods studies aimed at understanding barriers to or facilitators of the intervention. A systematic review of mixed-methods studies by Cardona-Arias et al. [33] pointed out that the use of mixed-methods designs is becoming increasingly common in evaluating the process, outcomes, effectiveness, and implementation of healthcare interventions, enhancing the understanding of causal mechanisms and process feedback, as well as the role of inter-subjective relationships in different interventions. Guided by the CFIR and RE-AIM frameworks and combined with the MRC guidelines, interviews with intervention staff and participants will be conducted to explore potential detailed factors in the implementation of the intervention from qualitative research, which will help to understand the opinions and suggestions of intervention staff on the implementation of the intervention and the views and suggestions of patients on the continued application of the intervention. At the same time, the parallel latent growth curve model will be applied to the quantitative exploration of the mechanisms of impact, and combined with the self-made goal achievement log book, it will further reflect the changes in patients' psychological measurements and the delivery of the intervention during the implementation process with objective data.

The resilience intervention embedded in this protocol is a patient-centered, individualized intervention, that is, targeted interventions are provided according to the different needs and individual circumstances of patients, thereby improving patients' adherence to the intervention and better helping patients enhance their resilience levels. Since the specific intervention content is provided according to the needs of patients, this protocol will reflect the intervention situation through quantitative data measured externally, while conducting interviews with different intervention subjects to understand the real and specific feelings of patients in different situations during the intervention and the common summaries and individual experiences of intervention staff for different intervention objects. The results of this protocol will help to adapt the resilience intervention to the conditions of different cancer patients and even other chronic disease patients and will enhance the sustainability of the intervention in practical applications.

The limitations of this protocol lie in the fact that the study only includes patients undergoing chemotherapy after GC surgery, without further considering the impact of factors such as GC classification, staging, and chemotherapy regimens, which may affect the reliability of the study results. Patients in the chemotherapy phase after GC surgery experience physiological stress from surgery and chemotherapy, as well as psychological distress such as anxiety and depression, and are more likely to develop severe psychological problems. The study focuses on this stage, aiming to help patients improve their psychological resilience through targeted psychological interventions. However, due to the limited sample size, further stratification of the patient population may result in subgroups that are too small, thereby affecting the power of statistical analyses and the reliability of the results. In future studies, the sample size will be expanded and more detailed stratifications will be conducted based on patients' different conditions, prognoses, and chemotherapy regimens in a broader population to enhance the representativeness and generalizability of the results. Additionally, due to ethical and practical reasons, the resilience intervention adopted a non-concurrent quasi-experimental design, with patients assigned to groups based on their admission order. This method does not achieve ideal randomization and may affect the scientific rigor and reliability of the study. However, by using admission time to assign patients who meet the inclusion and exclusion criteria to the control or intervention group, with a one-month washout period between the two groups to avoid temporal and spatial contamination, selection bias can be mitigated to some extent and the transparency and fairness of the grouping process can be ensured. This approach also avoids the ethical issue of including eligible patients in different groups within the same time frame. Moreover, grouping based on admission order ensures that the intervention can be implemented in a timely manner without delaying the intervention due to randomization waiting periods. Additionally, given the two-month intervention period, a blended approach combining online and offline methods is used to ensure continuous and complete delivery of the intervention content while reducing time costs. Therefore, in future studies, the research team will consider expanding the scope of intervention subjects, adopting a multi-center design, and combining randomization methods to further enhance the scientific rigor, reproducibility, and reliability of the study results.



## Supporting information

**S1 File. Goal achievement log book.**
(DOCX)

**S2 File. SPIRIT_Fillable-checklist.**
(DOC)

**S3 File. Interview guides.**
(DOCX)

**S4 File. Approved study protocol by ethics committee.**
(PDF)

**S5 File. Human_Subjects_Research_Checklist.**
(DOCX)

**S6 File. Ethics committee approval letter.**
(PDF)

**S7 File. Fund confirmation.**
(PDF)

## Acknowledgments

We would like to express our gratitude to the professionals, patients, and their families who participated in this study.

## Author contributions

**Conceptualization:** Xinqiong Zhang, Min Fan, Yanan Ding, Yuqin Pan, Zhang Xinqiong.

**Data curation:** Xinqiong Zhang, Min Fan.

**Formal analysis:** Xinqiong Zhang, Min Fan.

**Funding acquisition:** Yanan Ding, Zhang Xinqiong.

**Investigation:** Xinqiong Zhang, Min Fan, Yuqin Pan.

**Methodology:** Xinqiong Zhang, Yanan Ding, Yuqin Pan, Zhang Xinqiong.

**Project administration:** Yuqin Pan, Zhang Xinqiong.

**Resources:** Yuqin Pan, Zhang Xinqiong.

**Software:** Yanan Ding.

**Supervision:** Yuqin Pan, Zhang Xinqiong.

**Validation:** Xinqiong Zhang.

**Visualization:** Xinqiong Zhang, Min Fan.

**Writing – original draft:** Xinqiong Zhang.

**Writing – review & editing:** Xinqiong Zhang, Min Fan, Yanan Ding, Yuqin Pan, Zhang Xinqiong.

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
