## [Decision Letter · Decision Letter 0]

27 May 2025

PONE-D-24-38986Complex intervention based on protective factors to improve resilience for gastric cancer patients: mixed-methods process evaluation protocolPLOS ONE

Dear Dr. Xinqiong,

Thank you for submitting your manuscript to PLOS ONE. After careful consideration, we feel that it has merit but does not fully meet PLOS ONE’s publication criteria as it currently stands. Therefore, we invite you to submit a revised version of the manuscript that addresses the points raised during the review process.

We look forward to receiving your revised manuscript.

Kind regards,

Yu Uneno

Academic Editor

PLOS ONE

Journal Requirements:

“This paper was supported by the Humanities and Social Sciences Research Planning Fund of Ministry of Education of China (grant number: 17YJAZH126) and the Anhui Provincial Department of Education (grant number: yjs20210299).”

“This paper was supported by the Humanities and Social Sciences Research Planning Fund of Ministry of Education of China (grant number: 17YJAZH126) and the Anhui Provincial Department of Education (grant number: yjs20210299).”

Reviewers' comments:

Reviewer's Responses to Questions

**Comments to the Author**

1. Does the manuscript provide a valid rationale for the proposed study, with clearly identified and justified research questions?

Reviewer #1: Partly

Reviewer #2: Partly

Reviewer #3: Yes

Reviewer #4: Yes

Reviewer #5: Yes

Reviewer #6: No

2. Is the protocol technically sound and planned in a manner that will lead to a meaningful outcome and allow testing the stated hypotheses?

Reviewer #1: Partly

Reviewer #2: Partly

Reviewer #3: Partly

Reviewer #4: No

Reviewer #5: Yes

Reviewer #6: Partly

3. Is the methodology feasible and described in sufficient detail to allow the work to be replicable?

Reviewer #1: Yes

Reviewer #2: Yes

Reviewer #3: Yes

Reviewer #4: No

Reviewer #5: Yes

Reviewer #6: Yes

4. Have the authors described where all data underlying the findings will be made available when the study is complete?

Reviewer #1: Yes

Reviewer #2: Yes

Reviewer #3: Yes

Reviewer #4: Yes

Reviewer #5: Yes

Reviewer #6: Yes

5. Is the manuscript presented in an intelligible fashion and written in standard English?

Reviewer #1: Yes

Reviewer #2: No

Reviewer #3: No

Reviewer #4: Yes

Reviewer #5: Yes

Reviewer #6: No

6. Review Comments to the Author

You may also provide optional suggestions and comments to authors that they might find helpful in planning their study.

Reviewer #1: 1. This study provides a detailed description of a process evaluation protocol for a complex intervention. The protocol is designed to assess the effectiveness of the intervention and potentially be applied to the treatment of other diseases, which is a great idea. However, as a research study, the manuscript only presents the protocol itself, without providing evidence to verify whether this evaluation method is scientifically sound and efficient. It also lacks clarification on what kind of evaluation outcomes would indicate an effective or ineffective intervention.

2. It is recommended to explain the concept of psychological resilience in the introduction.

3. Since the full text of reference 11 could not be accessed(https://journals.lww.com/indianjcancer/pages/results.aspx?txtKeywords=An+intervention+based+onprotective+factors+to+improve++resiience+for+gastric+cancer+patients%3a+A+pilot+study), it is suggested to clarify in the discussion whether the study considered factors such as the classification and staging of gastric cancer, patients’ prognoses, and whether the chemotherapy regimens were consistent.

Reviewer #2: Dear Authors. I had the privilege to review your manuscript.

Despite aknowledging the vast amount and general quality of the work that you are planning, the protocol description has a number of flaws whcih prevents the reader to fully understand what you are planning to do.

First and foremost, in the actual form it is very difficult to understand what you are presenting. In the introduction you provide an extensive descritpion of a pilot study, to frame your protocol in an existing workflow. I would advise to move this part in a dedicate paragraph of the disucssion. Introduction should focus on framing your work in the actual body of evidence that fostering resilience helps pts with GC and so on. Some statements are also unclear to me, like those about "positive mindset" of patients. There is a body of literature on hope and resilience that should be accounted for. The method section is very detailed. Though you mention many times that this work is parte of MRC framework for complex interventions, it is unclear what the main stream of intervention is. Is this protocol a side work? Is it the core of the bigger intervention? Moreso, it in unclear whcih pts are you speaking about. Early GC pts? Advanced stage/terminal disease? I think this is also crucial, since hope and resilience change durign trajectory of disease and so do needs of pts.

The discussion is somehow repetitive of the description of the protocol that you provided in methods section. It should report your previosu reports and those of other authors on the matter and then, maybe, describe how your data and results will contribute. You may also comment on why you chose some framework or methods to collect and analyse data and not other ones.

The bibliography is updated but relatively "lean" for a complex work like this.

The overall flow of the paper is not very fluent and I would advise professional, English editing.

In conclusion, I would advise rewriting it in a simpler and more direct way, to celarly deliver your aims, methodology and expected results in the context of the research on resilience for pts with GC.

Reviewer #3: It is an interesting protocol. However, it could be improved with some corrections. My suggestions are:

Correct the first-person writing style and replace it with third-person.

Correct the location of the quotation numbering, as it does not comply with the Vancouver Standard.

Since this is a protocol for evaluating an intervention, the quantitative indicators should be more clearly defined (including their formula and expected threshold), as this will facilitate the analysis of the results.

It is suggested that the patient inclusion criteria be included.

It is suggested that details be provided on how the questionnaires assessing resilience, depression, social support, and other factors will be administered.

In the semi-structured interview guidelines, it is suggested that the targeted area be indicated (reach, efficacy, adoption, and maintenance); a table could be included given its importance.

Something very important is that reference No. 11 cannot be found. Furthermore, it is incomplete. This makes access to the results of the pilot study impossible.

There are other corrections that are highlighted in yellow with a note.

Reviewer #4: Thanks for inviting. However, the manuscript has several areas that require attention. Firstly, the study demonstrates limited novelty. Secondly, the description of intervention details is inadequate, as it neither explains how to prevent contamination among subjects due to interactions, nor how to avoid selection bias and standardize the research protocol. Thirdly, the manuscript fails to thoroughly detail the methods, steps, and scientific rationale of the intervention scheme, thus necessitating clarification on its development process, theoretical basis, and supportive evidence. Finally, there is a lack of a comprehensive discussion on quality control during the intervention process, covering aspects such as personnel, subject compliance, and data analysis, so the authors must provide a detailed account of quality control strategies for each stage to ensure the validity of the results and protocol.

Reviewer #5: Thank you for studying and disseminating your findings on the effect of a resilience intervention for gastric cancer patients. The integration of a psychological intervention to help with a physiological outcome is fascinating and using a mixed methods approach reflects this. A couple minor revisions I would suggest include clarifying sentence 2 in the introduction in which the authors describe severe psychological distress with examples and psychological challenges, without examples. I would recommend either providing examples of psychological challenges as the authors did with severe psychological distress, or simply provide definitions so the readers understand the difference. My last request would be for the authors to elaborate further on how participants were chosen for the control and intervention group. As its written, it appears, its based on when patients were admitted, which would be challenging for other researchers to replicate the study. Perhaps there were more criteria or some sort of additional randomization methodology that determined where participants were placed in the study; please clarify.

Reviewer #6: The submitted manuscript concerns an intervention aimed at strengthening resilience in patients with gastric cancer during chemotherapy. It goes beyond simply assessing the intervention's effectiveness, also seeking to understand the underlying mechanisms and processes involved. To achieve this, the study draws on established theoretical frameworks (MRC, CFIR, RE-AIM) and employs a mixed-methods design (qualitative and quantitative).

The study protocol incorporates the perspectives of both healthcare professionals and patients to better identify the factors that influence the implementation of the intervention. The expected outcomes may contribute to improving psychosocial care and promoting the development of more personalized support strategies.

In the introduction, the authors highlight the relevance of resilience evaluation and outline the project’s objectives and expected contributions. They state that the process evaluation will help refine resilience-based interventions for cancer patients, and potentially for individuals with other chronic illnesses.

The authors then detail the methodology that will be used to meet these objectives—a central component of the manuscript. In summary, the study aims to explore both the effects and internal functioning of a psychological intervention designed to support patients with gastric cancer. This will be achieved through a combination of interviews (exploring perceptions and experiences), quantitative measures (psychological and biological outcomes), and goal-tracking diaries (monitoring patients’ actual engagement).

Qualitative data will be collected via interviews with healthcare staff, to explore facilitators and barriers to implementation, and with patients, to understand their experiences and perceptions of the intervention. These interviews are structured using established frameworks (CFIR and RE-AIM), which guide the formulation of relevant questions to identify what works, for whom, and why.

Quantitative data will be gathered from patients through measures of resilience, hope, anxiety, depression, family and social functioning, as well as biological markers (e.g., C-reactive protein, cortisol). These assessments will be conducted at three time points, corresponding to each chemotherapy cycle, to track changes over time.

Patients will also complete goal-setting diaries throughout the intervention period. These diaries will be used to assess:

Fidelity (Did patients follow the planned intervention?),

Dose (How much time did they spend participating?),

Adherence (Did they complete the recommended activities?).

To ensure objectivity, the diaries will be analyzed by independent raters.

For data analysis, the authors plan to use specialized statistical models to track changes over time (e.g., latent growth curve modeling), with quantitative data analyzed using R software and qualitative data analyzed using NVivo.

The manuscript does not include a proper “Results” section. The authors write in the future tense, describing how the study will be conducted. They present the main objective of evaluating the implementation process of a resilience intervention in gastric cancer patients during chemotherapy, using a mixed-methods approach. They provide detailed descriptions of data collection methods, including semi-structured interviews and parallel latent growth curve models, as well as the goal-tracking diaries.

The strengths of the methodology are emphasized, including data triangulation (qualitative + quantitative), the use of robust theoretical frameworks (CFIR, RE-AIM), an in-depth analysis of mechanisms and change processes, and a patient- and provider-centered approach.

The discussion section outlines the expected contributions of the study and presents both its strengths and limitations.

However, the manuscript closely resembles a grant application rather than a research article suitable for publication in PLOS ONE. Supporting arguments include the exclusive use of the future tense in the "Results" section and the absence of any empirical data. For example, the discussion includes a paragraph stating: “To widely disseminate the research findings, we plan to submit the results of the process evaluation for publication in peer-reviewed journals and share them in scientific publications. We will also present our findings at academic conferences and seminars, engaging with healthcare professionals, psychologists, and nursing experts to promote the further development and application of the intervention.”

The overall writing style is dense, with numerous repetitions, making it difficult to follow—particularly for readers unfamiliar with research in the social and human sciences, including clinical psychology. For example, the following paragraph from the introduction illustrates the complexity of the writing: “Complex interventions share two common features: the complexity of the intervention itself and the complexity of its pathways. Process evaluation is a crucial component in designing and testing complex interventions, equally important as outcome evaluation. The resilience intervention comprises several interacting components and diverse intervention measures, contributing to its pathway complexity. Process evaluation is necessary to identify the mechanisms of each intervention component and the underlying logic behind the outcomes.”

The methodology may be difficult to grasp for researchers not already familiar with these types of intervention studies.

Given that the manuscript lacks actual results and primarily outlines a methodology—resembling a grant proposal—I do not recommend it for publication in PLOS ONE.

7. PLOS authors have the option to publish the peer review history of their article (what does this mean? ). If published, this will include your full peer review and any attached files.

**Do you want your identity to be public for this peer review?** For information about this choice, including consent withdrawal, please see our Privacy Policy .

Reviewer #1: No

Reviewer #2: No

Reviewer #3: No

Reviewer #4: No

Reviewer #5: **Yes: ** Sotera B. Chow, MA, MSN, RN, FNE-A

Reviewer #6: No

---

## [Author Response · Author response to Decision Letter 1]

11 Jul 2025

List of Responses

Dear Editors and Reviewers:

Thank you for your letter and for the reviewers’ comments concerning our manuscript entitled “Complex intervention based on protective factors to improve resilience for gastric cancer patients: mixed-methods process evaluation protocol” (ID: PONE-D-24-38986). Those comments are all valuable and very helpful for revising and improving our paper, as well as the important guiding significance to our researches. We have studied comments carefully and have made correction which we hope meet with approval. The responses to the comments of the reviewers are presented in blue font, and the revised text and the original modifications are underlined. In response to the suggestions of the reviewers, we have responded to each one below this letter.

Responds to the reviewer’s comments:

Reviewer #1:

Q1: The manuscript only presents the protocol itself, without providing evidence to verify whether this evaluation method is scientifically sound and efficient. It also lacks clarification on what kind of evaluation outcomes would indicate an effective or ineffective intervention.

Answer Thank you for your suggestion. In response to your query, the manuscript reports a process evaluation protocol embedded within a psychological resilience intervention study for gastric cancer patients undergoing chemotherapy. The purpose of this protocol is to explore the implementation process, mechanisms, influencing factors, and outcomes of the intervention, thereby providing a reference for the development of more comprehensive intervention strategies and broader application. Psychological resilience intervention for gastric cancer patients during chemotherapy is a complex intervention. Process evaluation is an essential component in designing and testing complex interventions, providing scientific evidence and empirical support for the successful implementation and continuous improvement of intervention measures. Therefore, we conducted this process evaluation study to understand the implementation process and the underlying logic of the outcomes. The process evaluation design of this study is guided by the MRC framework, the CFIR framework, and the RE-AIM framework. As reviewed in the relevant literature, the MRC framework is widely used for developing and evaluating complex interventions, the CFIR framework is commonly used to identify factors influencing the implementation of complex interventions, and the RE-AIM framework is one of the most widely used frameworks for planning and evaluating the implementation of complex interventions. Thus, this process evaluation study has a certain degree of scientific rigor and validity. Regarding the "evaluation results of the effectiveness of the intervention" you mentioned, this is the main content of the intervention effect evaluation study, which explores the effectiveness results through the measurement and comparison of outcome indicators before and after the intervention. Since this study is an introduction to the process evaluation protocol, it does not specify what kind of evaluation results indicate the effectiveness or ineffectiveness of the intervention. Thank you for your suggestion. To better clarify the manuscript, the following modifications have been made:

Resilience intervention is complex due to its multifaceted intervention needs across cognitive, emotional, behavioral, and social support dimensions, as well as the dynamic characteristics of individual differences, environmental factors, and psychological states. Therefore, resilience intervention is complex. Complex interventions share two common features: the complexity of the intervention itself and the complexity of its pathways[21]. Process evaluation is a crucial component in designing and testing complex interventions, equally important as outcome evaluation[22]. Process evaluation provides scientific evidence and empirical support for the successful implementation and continuous improvement of intervention measures from many aspects[21]. Therefore, resilience intervention needs to employ process evaluation to identify the mechanisms of each intervention component and the underlying logic behind the outcomes. This study elaborates on the process evaluation protocol for resilience intervention. The protocol aims to understand the implementation process and influencing factors, complementing the outcome evaluation by illustrating the impact of process evaluation on the results, thereby providing insights for developing more refined intervention strategies and broader application.

Q2: It is recommended to explain the concept of psychological resilience in the introduction.

Answer:Thank you for your recommendation. Based on your suggestion, I have added the following content to the introduction section of the manuscript:

Resilience is a process in which an individual, facing internal and external stress and adversity, activates latent internal cognitions, abilities, or psychological qualities and employs internal and external resources to actively repair and adjust, thereby acquiring the ability, process, or outcome of moving toward positive goals[10].

Q3: Since the full text of reference 11 could not be accessed, it is suggested to clarify in the discussion whether the study considered factors such as the classification and staging of gastric cancer, patients’prognoses, and whether the chemotherapy regimens were consistent.

Answer:Thank you very much for your valuable and professional suggestions. In response to your concern about the unavailability of the full text of reference 11, I would like to explain that this reference serves as the foundation for the current study. It primarily reports the preliminary application results of the psychological resilience intervention for gastric cancer patients undergoing chemotherapy in this study, demonstrating that the intervention has good feasibility and acceptability. This article is currently under submission, has been accepted and finalized, and is awaiting online publication.

In accordance with your suggestion, I have revised and clarified the discussion section of the manuscript as follows:

The limitations of this protocol lie in the fact that the study only includes patients undergoing chemotherapy after GC surgery, without further considering the impact of factors such as GC classification, staging, and chemotherapy regimens, which may affect the reliability of the study results. Patients in the chemotherapy phase after GC surgery experience physiological stress from surgery and chemotherapy, as well as psychological distress such as anxiety and depression, and are more likely to develop severe psychological problems. The study focuses on this stage, aiming to help patients improve their psychological resilience through targeted psychological interventions. However, due to the limited sample size, further stratification of the patient population may result in subgroups that are too small, thereby affecting the power of statistical analyses and the reliability of the results. In future studies, the sample size will be expanded and more detailed stratifications will be conducted based on patients' different conditions, prognoses, and chemotherapy regimens in a broader population to enhance the representativeness and generalizability of the results.

Reviewer #2:

Q1: In the introduction you provide an extensive descritpion of a pilot study, to frame your protocol in an existing workflow. I would advise to move this part in a dedicate paragraph of the disucssion. Introduction should focus on framing your work in the actual body of evidence that fostering resilience helps pts with GC and so on.Though you mention many times that this work is parte of MRC framework for complex interventions, it is unclear what the main stream of intervention is. Is this protocol a side work? Is it the core of the bigger intervention?

Answer Thank you very much for your questions and suggestions. The content of the introduction section has been supplemented and modified according to your suggestions, as follows:

The study by Zhang et al. found that individuals with higher resilience are usually able to evaluate themselves more positively and are more proactive in problem-solving and seeking help[12]. A systematic review of resilience in adult cancer patients by Tamura et al. revealed that resilience is closely related to alleviating anxiety and depression in cancer patients and improving their quality of life.

In response to your queries regarding the detailed introduction of the pilot study in the introduction section and the role and relationship of this protocol within the intervention, I would like to explain as follows: The manuscript reports a process evaluation protocol embedded within a psychological resilience intervention study for gastric cancer patients undergoing chemotherapy. The purpose of this protocol is to explore the implementation process, mechanisms, influencing factors, and outcomes of the intervention, thereby providing a reference for the development of more comprehensive intervention strategies and broader application. Therefore, the manuscript provides a detailed introduction to the intervention content in the introduction section as the research premise and basis for this protocol.

Moreover, the psychological resilience intervention for gastric cancer patients during chemotherapy is a complex intervention. Process evaluation is an essential component in designing and testing complex interventions and is equally important as outcome evaluation. It provides scientific evidence and empirical support for the successful implementation and continuous improvement of intervention measures from many aspects. Therefore, we conducted this process evaluation study to understand the implementation process and the underlying logic of the outcomes. The process evaluation design of this study is guided by the MRC framework, the CFIR framework, and the RE-AIM framework.

Thank you for your suggestion. To better clarify the manuscript, the following modifications have been made:

Resilience intervention is complex due to its multifaceted intervention needs across cognitive, emotional, behavioral, and social support dimensions, as well as the dynamic characteristics of individual differences, environmental factors, and psychological states. Therefore, resilience intervention is complex. Complex interventions share two common features: the complexity of the intervention itself and the complexity of its pathways[21]. Process evaluation is a crucial component in designing and testing complex interventions, equally important as outcome evaluation[22]. Process evaluation provides scientific evidence and empirical support for the successful implementation and continuous improvement of intervention measures from many aspects[21]. Therefore, resilience intervention needs to employ process evaluation to identify the mechanisms of each intervention component and the underlying logic behind the outcomes. This study elaborates on the process evaluation protocol for resilience intervention. The protocol aims to understand the implementation process and influencing factors, complementing the outcome evaluation by illustrating the impact of process evaluation on the results, thereby providing insights for developing more refined intervention strategies and broader application.

Q2: Some statements are also unclear to me, like those about "positive mindset" of patients. There is a body of literature on hope and resilience that should be accounted for.

Answer Thank you very much for your suggestions. These suggestions can help to further refine the content of the manuscript. In accordance with your suggestions, the concept of psychological resilience and related literature have been supplemented in the manuscript, which can help to understand the expression "positive mindset" of patients. Regarding "hope," it is mentioned in the manuscript as one of the protective factors of psychological resilience. Together with other protective factors, it has been incorporated into the development of the intervention program as a key consideration. Since this manuscript is a report on the process evaluation protocol embedded in the intervention, the introduction section does not provide an extensive discussion. The specific modifications and supplements are as follows:

Resilience is a process in which an individual, facing internal and external stress and adversity, activates latent internal cognitions, abilities, or psychological qualities and employs internal and external resources to actively repair and adjust, thereby acquiring the ability, process, or outcome of moving toward positive goals[10].

The study by Zhang et al. found that individuals with higher resilience are usually able to evaluate themselves more positively and are more proactive in problem-solving and seeking help[12]. A systematic review of resilience in adult cancer patients by Tamura et al. revealed that resilience is closely related to alleviating anxiety and depression in cancer patients and improving their quality of life.

Q3:It in unclear whcih pts are you speaking about. Early GC pts? Advanced stage/terminal disease?

Answer Thank you very much for your question. Your query is highly targeted and can help further refine the content of the manuscript. In response to the issue of unclear staging of gastric cancer patients, the discussion section of the manuscript has been supplemented accordingly. The specific content is as follows:

The limitations of this protocol lie in the fact that the study only includes patients undergoing chemotherapy after GC surgery, without further considering the impact of factors such as GC classification, staging, and chemotherapy regimens, which may affect the reliability of the study results. Patients in the chemotherapy phase after GC surgery experience physiological stress from surgery and chemotherapy, as well as psychological distress such as anxiety and depression, and are more likely to develop severe psychological problems. The study focuses on this stage, aiming to help patients improve their psychological resilience through targeted psychological interventions. However, due to the limited sample size, further stratification of the patient population may result in subgroups that are too small, thereby affecting the power of statistical analyses and the reliability of the results. In future studies, the sample size will be expanded and more detailed stratifications will be conducted based on patients' different conditions, prognoses, and chemotherapy regimens in a broader population to enhance the representativeness and generalizability of the results.

Q4:The discussion is somehow repetitive of the description of the protocol that you provided in methods section. It should report your previosu reports and those of other authors on the matter and then, maybe, describe how your data and results will contribute. You may also comment on why you chose some framework or methods to collect and analyse data and not other ones.

Answer Thank you very much for your suggestions. The discussion section has been modified accordingly. The specific modifications are as follows:

This study outlines a process evaluation protocol for a complex intervention aimed at enhancing resilience in GC patients during chemotherapy. The protocol explores the implementation process of the intervention to complement and substantiate the results of the effectiveness evaluation. It also revises and refines the intervention program to form a more comprehensive intervention strategy, thereby facilitating the broader application of the intervention among other chronic disease patient populations. Guided by the MRC guidelines[21], CFIR framework[25], and RE-AIM framework[26], the protocol ensures the scientific rigor and completeness of the study. The MRC guidelines are widely used for the development and evaluation of complex interventions. A study by Collet et al.[29] showed that the CFIR framework is a widely used framework to identify facilitators and barriers to the successful implementation of healthcare interventions. A systematic review of RE-AIM utilization by D'Lima et al.[30] indicated that the RE-AIM

---

## [Editor Report · Decision Letter 1]

23 Jul 2025

Complex intervention based on protective factors to improve resilience for gastric cancer patients: mixed-methods process evaluation protocol

PONE-D-24-38986R1

Dear Dr. Xinqiong,

We’re pleased to inform you that your manuscript has been judged scientifically suitable for publication and will be formally accepted for publication once it meets all outstanding technical requirements.

Kind regards,

Yu Uneno

Academic Editor

PLOS ONE
---

## [Editor Report · Acceptance letter]

PONE-D-24-38986R1

PLOS ONE

Dear Dr. Xinqiong,

I'm pleased to inform you that your manuscript has been deemed suitable for publication in PLOS ONE. Congratulations! Your manuscript is now being handed over to our production team.

Kind regards,

on behalf of

Dr. Yu Uneno

Academic Editor

PLOS ONE